

# 1 Impacts of Long-range Transport of Aerosols on Marine Boundary Layer Clouds in
# 2 the Eastern North Atlantic

**Yuan Wang[1,2,*], Xiaojian Zheng[3], Xiquan Dong[3], Baike Xi[3], Peng Wu[3], Timothy Logan[4],**
**Yuk L. Yung[1,2]**
[1]Division of Geological and Planetary Sciences, California Institute of Technology, Pasadena,
CA, USA
[2]Jet Propulsion Laboratory, California Institute of Technology, Pasadena, CA, USA
[3]Department of Hydrology and Atmospheric Sciences, University of Arizona, Tucson, AZ, USA
[4]Department of Atmospheric Sciences, Texas A&M University, College Station, TX, USA
*Corresponding author: Yuan Wang (yuan.wang@caltech.edu)





**Abstract**
Vertical profiles of aerosols are inadequately observed and poorly represented in climate models,
contributing to the current large uncertainty associated with aerosol-cloud interactions. The DOE
ARM Aerosol and Cloud Experiments in the Eastern North Atlantic (ACE-ENA) aircraft field
campaign near the Azores islands provided ample accurate observations of vertical distributions
of aerosol and cloud properties. Here we utilize the in situ aircraft measurements from the ACE-
ENA and ground-based remote sensing data along with an aerosol-aware Weather Research and
Forecast (WRF) model to characterize the aerosols due to long-range transport over a remote
region and to assess their possible influence on marine boundary-layer (MBL) clouds. The vertical
profiles of aerosol and cloud properties measured via aircraft during the ACE-ENA campaign
provide detailed information revealing the physical contact between transported aerosols and MBL
clouds. The ECMWF-CAMS aerosol reanalysis data can reproduce the key features of aerosol
vertical profiles in the remote region. The cloud-resolving WRF sensitivity experiments with
distinctive aerosol profiles suggest that the transported aerosols and MBL cloud interactions (ACI)
require not only low-altitude aerosol preferably getting close to the marine boundary layer top, but
also large cloud top height variations. Based on those criteria, the observations show the
occurrence of ACI involving the transport of aerosol over the Eastern North Atlantic is about 62%
in summer. For the case with noticeable long-range transport aerosol effect on MBL cloud, the
susceptibilities of droplet effective radius and liquid water content are $-0.11$ and $+0.14$,
respectively. When varying on the similar magnitude, aerosols originating from the boundary layer
exert larger microphysical influence on MBL clouds than those entrained from free troposphere.





## 1. Motivation and Background

It has been long hypothesized that increased high concentrations of aerosols serving as cloud condensation nuclei (CCN) can reduce cloud droplet effective radius, enhance cloud albedo, suppress drizzle formation, and change cloud lifetime and fraction, the so-called aerosol indirect effects (AIE) (Twomey, 1977; Seinfeld et al., 2016). However, current radiative forcing stemming from cloud responses to anthropogenic aerosols remains highly uncertain in the climate system, representing the largest challenge in climate predictions (Fan et al., 2016). Note that the current IPCC assessment mainly considers the warm stratus and stratocumulus response to aerosols (Myhre et al., IPCC, 2013), while aerosol induced convective cloud response (Wang et al., 2014) as well as with anthropogenic aerosol effect as ice nuclei (Zhao et al., 2019) have not been fully accounted for yet. Even for warm clouds, the climate significance of whether liquid water content and cloud lifetime are enhanced or reduced by CCN is still widely debated (Malavelle et al., 2017; Toll et al., 2019; Rosenfeld et al., 2019). Due to the nonlinear nature of cloud responses to CCN perturbations, the largest cloud susceptibility and AIE typically occurs for marine boundary layer (MBL) clouds over remote regions (Garrett and Hobbs, 1995; Carslaw et al., 2014; Dong et al., 2015). Under the pristine conditions with extremely low background CCN concentration (Kristensen et al., 2016), any aerosol intrusion following long-range transport has great potential to alter the local aerosol/CCN budget (Roberts et al., 2006). Hence, in this study, we aim to characterize long-range transport of aerosols and to assess their impacts on MBL clouds by combining in situ aircraft measurements with cloud-resolving model simulations.

For those aerosols resulting from long-range transport, one of the most important aspects pertinent to aerosol-cloud interactions (ACI) is their vertical distribution, or in other words, their position relative to cloud layers. The vertical distribution of aerosols can be affected by a number of complex atmospheric processes, such as emission, transport, deposition, as well as microphysical and chemical processes. Previous studies suggest that aerosols can alter MBL cloud microphysical properties and enhance indirect effects through entrainment into the cloud top when either aerosol particles settle or the cloud deck deepens (Painemal et al., 2014, Lu et al., 2018). In the boundary layer of remote regions like the equatorial Pacific, the majority of CCN were found to be supplied by long-range transport instead of local emission or formation (Clarke et al., 2013). Recent aircraft observations from the NASA's Ob-seRvations of Aerosols above CLouds and their intEractionS (ORACLES) campaign showed distinctive MBL cloud responses to aerosols above





and below cloud depending on the history of smoke entrainment (Diamond et al., 2018). Therefore,
it is critical to understand aerosol variability as a function of height and its influence on the aerosol
indirect forcing assessment over the regions where MBL clouds are abundant.

Spaceborne active sensors that possess vertically profiling capability have been widely

used to characterize aerosol and cloud spatial variations and to detect the aerosol above clouds
(Painemal et al., 2014; Jiang et al., 2018). However, satellites likely miss the thin aerosol layers
with relatively low concentration (but still higher than maritime background values), and thus
overestimate the distance between the aerosol plume base and the cloud top using the spaceborne
observations. Therefore, aircraft observations with continuous vertical sampling are the most
reliable source that can accurately characterize the vertical relationship between aerosol and cloud.
The DOE ARM Aerosol and Cloud Experiments in the Eastern North Atlantic (ACE-ENA) aircraft
field campaign near the Azores islands provided a unique opportunity to study aerosols from
different sources and their impacts on MBL clouds (Wang et al., 2019). The ENA site is located
in the remote northeastern Atlantic Ocean where MBL clouds are prevalent throughout the year
due to the warm sea surface temperature and prevailing subsidence near the edge of the Hadley
cell (Wood et al., 2015, Dong et al. 2014). The site also receives complex air mass dictated by
different wind patterns. In addition to the local maritime air, the airflows originating from either
the North American or the Saharan region complicate the local aerosol types and sources (Logan
et al., 2014).   This study leverages the airborne measurements of aerosol vertical profiles for
different chemical species to understand aerosols and their influence on MBL cloud microphysical
properties over the Azores, with the ultimate goal to provide observational constraints on the global
climate model simulations. An aerosol reanalysis product is evaluated in the present study as well.

Even with aircraft measured vertical relationship between aerosol and cloud, it is difficult

to estimate whether the aerosol aloft can impact the cloud beneath, as the microphysical processes
such as entrainment into cloud top cannot be directly measured. Hence, we employ aerosol-aware
cloud-resolving simulations to simulate the MBL cloud development and aerosol transport in the
free troposphere and to quantify the AIE. Through the sensitivity experiment by imposing different
aerosol vertical profiles, we can disentangle aerosol and other confounding meteorological factors
in ACI, which is challenging to do using only short-term observations. Section 2 describes the
main observational data and introduces the numerical modeling tools. Section 3 reports the
observed aerosols and clouds based on aircraft measurements and reanalysis product. Section 4





presents the analyses of cloud-resolving simulations using the WRF model. Section 5 summarizes the key finding in this study and provide additional discussions for the study's caveats and future work.

**2. Methodology**

**2.1 Aircraft Observations and Ancillary Data Descriptions**

Vertical distributions of aerosols and MBL cloud microphysical properties over the Azores were obtained during ACE-ENA two intensive operational periods (IOPs), i.e. early summer 2017 (late June to July) and winter 2018 (January to February). Since the aerosol concentration and variability are much larger in the summertime of Azores, we will mainly focus on the 2017 July in this study. The ARM Aerial Facility (AAF) Gulfstream-159 (G-1) provides accurate measurements of aerosol size distribution, total aerosol number concentration, and chemical constituents below and above cloud layers during the summer IOP. The Condensation Particle Counter (CPC) on board the G1 can detect aerosol particles larger than 10 nm, and it can provide profiles of condensation nuclei number concentration ($N_{CN}$) when the aircraft ascends or descends. Note that $N_{CN}$ measurements inside cloud can be contaminated and thus have large uncertainty. Cloud condensation nuclei (CCN) number concentration ($N_{CCN}$) is obtained by the CCN-200 particle counter on board the G1 aircraft. The $N_{CCN}$ is measurement under the controlled supersaturation of 0.35% with the humidified particle size range from 0.75 μm to 10 μm (Rose et al., 2008). We analyze sulfate and organic carbon (OC) mass concentrations measured by the Aerodyne high-resolution time of flight aerosol mass spectrometer (HR-ToF-AMS) and refractory black carbon (BC) from the Single Particle Soot Photometer (SP2).

We use cloud and drizzle microphysical property profiles retrieved from a combination of ground-based observations including a Ka-band ARM Zenith Radar, ceilometer, and microwave radiometer. Fast Cloud Droplet Probe (Glienke and Mei, 2020) measured cloud droplet properties (diameter between 1.5 and 46 μm), and 2-Dimensional Stereo Prob (2DS, Glienke and Mei, 2019) measured drizzle properties (diameter greater than 45 μm) were used to evaluate the ground-based retrievals. Following Dong et al. (1997) and Frisch et al. (1995, 1998), cloud droplet size distribution was assumed as a lognormal distribution. Differently, drizzle size distribution was assumed as a normalized Gamma distribution, as suggested by O'Connor et al. (2005) and Ulbrich (1983). The retrieved cloud and drizzle properties are validated against collocated aircraft in situ measurements during ACE-ENA (Wu et al., 2020). Both the time series and vertical profiles from





the retrievals agree well with in situ observations. Treating the aircraft measurements as cloud
truth, the median retrieval uncertainties are estimated as ~20% for cloud droplet effective radius,
~30% for cloud droplet number concentration, liquid water content (LWC) and drizzle drop
median radius.
To characterize long-range aerosol intrusions over the monthly time scale, we employ
global aerosol reanalysis data, namely the Copernicus Atmosphere Monitoring Service (CAMS).
It provides four-dimensional mass concentrations of aerosols and reactive gases with a horizontal
spatial resolution of approximately 80 km and 60 vertical levels. The CAMS reanalysis was
constructed by assimilating several satellite products of the atmospheric constituents into a global
model and data assimilation system (Flemming et al., 2017). The assimilated satellite datasets
include aerosol optical depth (AOD) from MODIS and AATSR, CO from MOPITT, $NO_2$ and $O_3$
from OMI, GOMES, etc.
**2.2 Model Description**
The Weather Research and Forecasting (WRF) model version 3.6 is employed in this study
to simulate MBL clouds and their possible interactions with transported aerosols. Four nested
domains are setup with horizontal resolutions of 19.2 km, 4.8 km, 1.2 km, and 300 m (SI Fig. 1).
Even for the innermost domain, we try to cover as large area as possible, considering the highly
heterogeneous meteorological conditions in the mid-latitudes. The innermost domain is configured
in a similar way with large-eddy simulations and it uses the 3-dimensional Smagorinsky first order
closure for eddy coefficient computation. Boundary layer parameterization is turned off for this
domain. Note that 300-m horizontal resolution does not strictly meet the classic LES requirement,
but recent simulations with similar resolutions successfully reproduced the structure and drizzle
onset of MBL clouds (Wang and Feingold, 2009) and were used to study boundary layer cloud
interactions with aerosols (Lin et al., 2016). The 65 stretched sigma levels are used with a 40 m
vertical resolution within MBL. The large-scale forcing is adopted from the ERA5 reanalysis data
with 25 km horizontal resolution (Copernicus Climate Change Service, 2017).
To accurately depict MBL cloud microphysical processes, a spectral bin microphysical
(SBM) scheme is employed which utilizes a pair of 33 bins to represent cloud/rain drops and
aerosols separately without prescribed size distributions (Fan et al., 2012; Wang et al., 2013).
Aerosol activation is explicitly calculated using the model predicted water vapor supersaturation.
The Kölher theory is used to calculate the critical radius. The hygroscopicity of sulfate is assumed





for aerosols in each size bin. At each timestep, aerosols with radius greater than the critical radius
are removed from the aerosol spectrum and the mass of the activated droplets is added to the cloud
spectrum. Aerosol regeneration from complete evaporation of droplets and/or raindrops is also
considered in SBM. Since the aerosol size distribution in SBM ranges from a few nanometers to a
few microns, the definition of aerosol in the model is closer to the condensation nuclei in the
aircraft observation. Hence, observed vertical profiles of $N_{CN}$ from selected cases are used for the
initial and lateral boundary conditions of aerosols in the model. The model integrates from 1200
UTC on the day before the selected case, and the first 12 hours is considered as spin-up. Shortwave
and longwave radiation transfer calculations are accounted for by the Goddard and RRTM schemes,
respectively. The radiative effect of aerosols above the cloud decks is not considered in the present
model setup. We speculate such an effect is small, because of rather low aerosol optical depth over
this remote region, even with the long-range transported aerosols (aside from thick dust plumes
from the Saharan Desert).
**3. Observational Data Analysis**
**3.1 Characterization of aerosol vertical distribution using the CAMS reanalysis**

Previous study showed that the CAMS aerosol product exhibit good agreement with

ground-based observations such as AERONET and unassimilated satellite products such as MISR
on the global scale (Christophe et al., 2019). The global spatial correlation of CAMS AOD with
AERONET is about 0.83, and the bias in CAMS AOD seasonal variation is between -10% and
+20%. Here we utilize this dataset to characterize the aerosol vertical distribution over the
northeast Atlantic during the ACE-ENA field campaign. Vertical distributions and their temporal
evolutions for five types of aerosols, including sulfate, organic carbon (OC), black carbon (BC),
sea salt, and dust, over the whole month of July 2017 are displayed in Fig. 1 based on the CAMS
aerosol reanalysis. Sulfate, OC, and BC are the predominant aerosol types possibly possessing an
anthropogenic signature. BC and OC can also originate from biomass burning. Those aerosols
share a similar spatiotemporal pattern in the free troposphere, indicating that they undergo similar
long-range transport before arriving over the Azores island. Marked and persistent low-altitude (1-
2 km) pollution transport occurred between 1-13 July, as shown in the evolution of vertical profiles
of sulfate, OC and, BC (Figs. 1a-1c). High-altitude (3-6 km) pollution transport occurred between
6-20 July for those three aerosol types as well. Both modes of pollution transport occurred 50% of
the time during July 2017, indicating a high frequency of long-range transport over this area. July


18 and 12 presents the typical high- and low- plume cases, respectively, so they will be investigated
thoroughly in the later aircraft data analyses and model simulations. The concentrations of OC,
BC, and sulfate are generally low in the MBL, so aerosol penetration from the free troposphere
into the lower MBL may be not significant during this month. One exception is sulfate during 18-
21 July. Sulfate concentration experienced an increase in the MBL followed by a lag increase in
the free troposphere. Since there is no significant transport signal before and during that time
period, the elevated sulfate concentration within the boundary layer is due likely to some local
sources such as oxidation of marine dimethyl sulfate (DMS).

The aerosols of natural sources, namely sea salt and dust, show different vertical

distributions (Figs. 1d -1e). Sea salt aerosols mainly reside near the surface and are rarely found
above 1000 m. Dust particles are mainly found at high altitudes, typically above 3 km, during 5-
14 July, indicating their long-range transport. However, the dust spatiotemporal pattern in the free
troposphere are quite distinctive from sulfate and smoke, implying the different sources of long-
range transport. Previous studies suggest the possible dust transport from the Saharan Desert to
the northeast Atlantic region (Logan et al., 2014; Weinzierl et al., 2015). To address those issues,
back-trajectory analyses were conducted, and the results will be discussed later. During 15-19 July,
dust particles are found within the boundary layer and even near the surface following the presence
of dust plume in the free troposphere earlier. Such a downward propagation does not occur for
anthropogenic aerosols either, likely explained by the fact that dust particles are bigger in size with
larger settling velocity.
**3.2 Identification of source regions using back-trajectory analysis**

The backward ensemble trajectories were computed using the NOAA Hybrid Single-

Particle Lagrangian Integrated Trajectory (HYSPLIT) (Stein et al., 2015) model, based on the
large-scale meteorological fields from Global Data Assimilation System (GDAS) with a spatial
resolution of 0.5°. We focus on three cases/days to examine the sources of typical high- and low-
altitude plumes of anthropogenic aerosols and mineral dust. The model uses an end-point height
of 1.5, 2.4, and 3 km for three selected cases to represent the air parcels in the anthropogenic low-
altitude, high-altitude, and dust plumes, respectively. To capture the different lengths of transport
procedure, the model was backward integrated for 7 days for the anthropogenic aerosols and 13
days for the mineral dust case. 20 ensemble members are employed for each case. They agree with
each other better on horizontal trajectory than vertical displacement. Larger differences are found



among the ensemble members after three days for anthropogenic aerosols and after two days for
dust.
The back-trajectory analyses confirm that the source region of sulfate, BC, and OC in the
plumes is the North American continent (Fig. 2a,c), consistent with previous analyses of data from
the earlier field campaign over the ENA site (Logan et al., 2014). The westerly jet carries the
pollutants across the Atlantic Ocean, and it takes three to four days to arrive the Azores. Temporal
evolutions of trajectory vertical displacement reveal when aerosols are elevated from the PBL to
the free troposphere and such information can be used to pinpoint the aerosol source. Fig. 2b,d
suggests that aerosols are mainly from the central US in the high-plume case, and from eastern US
in the low-plume case. The curved trajectories in the low-plume case reflect the influence of the
Bermuda/Azores High located to the south. The dust transports exhibit a much different pathway.
Starting at 3km altitude, the back-trajectory develops westward initially, but sharply turn around
and point to the North Africa (Fig. 2e,f). It suggests that Sahara is the most likely source for the
dust particles observed over the Azores.
Note that back-trajectory analysis of air mass has its own limitations. For example,
shipping emissions over Northern Atlantic Ocean are not considered in the present analysis. Also,
the source attribution based on episodic events may be not representative for the climatological
mean scenario. Therefore, the source attribution results here need to be further evaluated in future
studies which can utilize more sophisticated approach such as source tagging in the GCM nudged
by the reanalysis data (Wang et al., 2014).
**3.3 Vertical distributions of different aerosols in aircraft observations**
Aircraft observations during the ACE-ENA provide more accurate depictions of aerosol
vertical distribution and aerosol layer heights relative to cloud layer heights, with differentiation
of aerosols type and hygroscopicity. During the summer IOP, quite diverse aerosol vertical profiles
are found. Here we focus on those with noticeable aerosol plumes in the free troposphere. Fig. 3
shows two representative vertical distributions of aerosol mass concentrations averaged over the
flights on July 18 and 12, corresponding to the high- and low-altitude aerosol plume, respectively.
In the high-altitude plume case, BC, OC, and sulfate concentrations all increase with height above
clouds, indicating downward propagation of aerosol plumes and possible interaction with MBL
clouds. BC and OC concentrations are even higher than that of sulfate in the free troposphere,
suggesting the biomass burning signature of the plume on that day. Conversely, within MBL, much





higher concentration of sulfate in the MBL than those of BC and OC. This phenomenon is also
captured by the CAMS aerosol reanalysis (Fig. 1a), lending support to the fidelity of the reanalysis
dataset. For the low altitude plume (Fig. 3b), the vertical gradients of aerosol concentrations are
not clear above clouds, but aerosol concentrations within 500 m right above clouds are higher than
those near the cloud base (Fig. 3b), corroborating the physical contact between aerosol plumes and
MBL clouds. Comparing Fig. 3 and 1, the CAMS reanalysis data generally agree with aircraft
observed aerosol profiles on the selected days, but the predicted aerosol mass mixing ratios are an
order of magnitude higher in the reanalysis data. Those discrepancies point out that any
quantitative usage of aerosol reanalysis product should be cautious.

Aerosol and CCN concentration vertical profiles are also available from the aircraft

observations. For the high-altitude plume, $N_{CN}$ reaches a peak of ~ 600 cm$^{-3}$ at 2.5 km, and then
decreases dramatically downwards to ~180 cm$^{-3}$ near cloud top (~ 1.1 km), which is even lower
than $N_{CN}$ values within the boundary layer ranging from 200 to 300 cm$^{-3}$ on that day (Figure 4a).
The measured 200-m average of $N_{CN}$ above cloud top is 185 cm$^{-3}$, smaller than that below cloud
base 290 cm$^{-3}$ (Table 1). From the surface to the 2.5 km height, the minimum $N_{CN}$ occurs near
cloud top, reflecting the disconnection between MBL aerosols and those from long-range transport
aloft. The characteristics of $N_{CCN}$ profile are similar with those of $N_{CN}$. In the low-altitude plume,
both $N_{CN}$ and $N_{CCN}$ show a slower decline of above the cloud layer (Fig. 4c,d). Also, the right-
above-cloud-top $N_{CN}$ and $N_{CCN}$ at 1 km are higher than those below the cloud layer, indicating the
physical contact of the aerosol plume with the cloud deck.

During the summer IOP, the aircraft was deployed in twenty days to collect data. Among

those days, only eight of them have stable MBL clouds during the flight hours, according to the
ground-based cloud radar. We summarize the aircraft observed aerosol and cloud vertical
distribution characterizations for those eight days/cases in Table 1. Among those eight cases, five
days show an increase in above-cloud $N_{CN}$ along with height, and one day shows roughly constant
$N_{CN}$ above clouds, all of which indicate the existence of long-range transport of aerosols in the
free troposphere and downward propagating influence on the aerosol budget near the cloud top.
Moreover, five out of eight cases have above-cloud $N_{CN}$ (within 200 m) significantly larger than
below cloud $N_{CN}$, implying the potential influence of free-troposphere aerosols on MBL clouds
from another angle of view.
**4. WRF modeling of MBL clouds and their response to transported aerosols**



In observation of quite diverse aerosol vertical profiles in the real atmosphere, an
outstanding science question is under what conditions the long-range transported aerosols can
exert significant impacts on the MBL clouds beneath. To answer this question and to quantify the
related aerosol indirect effects, cloud-resolving WRF simulations are performed, focusing on the
two selected cases with the high- and low-altitude plume on 18 July and 11 July, respectively. In
the model control simulations, the aircraft measured aerosol profiles are used to set up initial and
lateral boundary conditions of aerosol total number concentration for the two cases (Fig. 5).
Sensitivity simulations for clean scenarios are conducted by replacing the observed aerosol
concentrations above cloud with an assumed exponential decrease of $N_{CN}$ along with height in the
free troposphere instead. Before sensitivity analyses, we want to examine to what extent the cloud-
resolving simulations can reproduce the local-scale meteorological variations and MBL cloud
structure at Azores. Here we use the high-altitude plume case as an example to evaluate the
model's fidelity in the northeast Atlantic.
The large-scale wind pattern and boundary layer structure from the model control run are
compared against the interpolated soundings over the ARM ENA site. Fig. 6 shows that the model
exhibits good agreement with the observed air temperature, moisture content, and relative
humidity. The model captures the cold/dry air advection at 1 km height in the morning followed
by the warm/moist air in the afternoon. The persistent supersaturation between 500 and 1000 m
and associated cloud deck are also reproduced in the simulation. We find that the key model
configuration to reproduce the main features of meteorological variability is to have appropriate
domain nesting and dynamical downscaling. Particularly, the outmost domain with 19.2 km grid
spacing is crucial and necessary for this mid-latitude region. The region is featured by frequent
mesoscale weather systems, and local wind and moisture fields vary drastically even within a day.
The model setup with only three domains of 4.8 km, 1.2 km, and 300 m horizontal resolution
induce large errors in the vertical profiles of moisture and temperature. A persistent dry bias occurs
near the MBL top when the outmost domain with 19.2 km grid spacing is absent. Such
meteorological biases further influence cloud simulation and result in discontinuous cloud layer in
its temporal evolution.
MBL cloud properties simulated by WRF are evaluated against the retrievals from a
combination of ground-based observations. The simulation captures the cloud top height at 1km
and cloud bottom height at 500 m during the day (Fig. 7a,b). Therefore, the cloud physical





thickness is comparable between model and observation. LWC is generally smaller in the model
than that in the observation. Meanwhile, the simulation captures the larger LWC near the top of
the cloud, reflecting the adiabatic growth of cloud droplet starting from the cloud bottom. The
temporal evolution of simulated LWCs does not match well with retrievals, partly due to the spatial
sampling bias. Cloud droplet effective radius ($R_e$) in the model is calculated as a function of
volume-mean droplet radius as well as relative dispersion (a ratio between standard deviation and
mean radius in a size distribution) (Liu and Daum, 2002). The model shows the comparable
vertical distribution of $R_e$ with cloud radar retrievals, e.g. the larger $R_e$ near the cloud top, but with
larger variability in the size range than observations (Fig. 7c,d).
To explore the sensitivity of MBL cloud microphysical properties to the long-range aerosol
transport, we contrast the simulations with and without observed long-range aerosol plumes in the
free troposphere. For the high-altitude plume (July 18) case, the comparisons of model run with
different aerosol vertical profiles show that both LWC and cloud fraction remain largely
unchanged, whether the aerosol plume above 1.5 km exists or not. In fact, the cloud top height on
that day experienced some temporal variations near the Azores, as it extended to 1.5 km during
the night due to strong radiative cooling and reduced to 1 km during the most of daytime. As a
result, the distance between the aerosol plume and cloud deck varied from 500 m to less than 100
m. Fig. 8a-f show that the long-range transported aerosols have no significant impacts on the MBL
cloud properties underneath when the physical distance between aerosol plume and cloud layer is
greater than 100 m. This finding does not support the previous study based on satellite products
arguing that aerosol-cloud interactions are still discernable with aerosol plumes 1 km above the
cloud deck (Painemal et al. 2014).
To answer the question at what height aerosol plume starts to influence MBL cloud
microphysical properties, we perform an additional simulation by lowering the aerosol plume
bottom from 1.5 km to 1.1 km which is considered as the height of MBL and cloud tops during
the daytime. In this sensitivity run, the aerosol indirect effect remains largely muted during the
daytime. It suggests that when boundary layers and cloud decks are relatively stable, long-range
transport aerosols have a low chance of being entrained into the cloud top and being activated to
cloud droplets. However, when the cloud deck becomes deeper at night, particularly after 2200
UTC when a significant part of the cloud extends into the aerosol layer above 1.1 km, an increase
in LWC by up to 0.1 g m$^{-3}$ is observed (Fig. 8g-h).





In contrast, the simulated clouds in the low-altitude plume (July 12) case exhibit large
variations in the vertical (Fig. 9), and consequently the aerosol plume just above the cloud top
imposes significant influence on the MBL cloud micro- and macro-physical properties. The mean
LWC is increased by 5.7%, and cloud fraction is increased by 5.4%, due to a 48.0% increase in
CCN under the influence of the long-range aerosol transport. The distinctive responses of MBL
clouds to aerosol plumes at different heights reinforce the notion that the vertical overlap between
aerosol and cloud layers is crucial for ACI pertinent to the long-range aerosol transport. Moreover,
the extent of overlap is jointly controlled by aerosol plume height and cloud top variation. The
latter is particularly important, when the boundary layer is relatively stable, and the aerosol vertical
mixing is rather weak for most marine stratus.
It is a nontrivial task to identify the physical contact between an aerosol plume and a cloud
deck based on the aircraft measurements. Especially when the center of an aerosol plume is
hundreds of meters above cloud top and aerosol concentration right above the cloud is lower than
that within PBL, it is difficult to estimate whether aerosols can be entrained into the cloud layer.
As the above model results suggested, ACI requires critical mass of aerosols immersed into the
cloud layers. Here we define a "critical altitude" at which above-cloud $N_{CN}$ is equal to the below-
cloud $N_{CN}$. With such a concept, we can compare this altitude to the cloud top variation during a
period of interest. Take the July 18 case for example, according to the airborne measurements, the
critical altitude is 1674 m, well beyond the range of cloud top variation (880 – 1300 m) on that
day (Table 1). Thus, we can reach a conclusion that, even though long transport of aerosols was
found in the free troposphere on that day, they were unlikely to interfere with MBL clouds below.
Here we take all the airborne measured vertical information into account, including aerosol
changes above clouds, comparison of above- and below-cloud $N_{CN}$, as well as cloud top height
variations, and We revisit the eight observed cases in Table 1. We find that five days (0628, 0630,
0706, 0712, and 0715) out of eight during the summer phase of the ACE-ENA field campaign
clearly show the interactions between aerosols from long-range transport and local MBL clouds,
corresponding to a 62.5% occurrence frequency.
The previous cloud-resolving modeling studies of aerosol effects on MBL cloud properties
either used a constant CCN concentration throughout the whole domain (Yamaguchi et al., 2019)
or the CCN profiles in MBL were prescribed with an exponential decrease in the free troposphere
(Wang et al., 2013, 2018; Lin et al., 2016). The consequent sensitivity experiments were conducted





by perturbing CCN at different heights with the same scaling factor, without differentiating the
aerosols from different sources. Therefore, those studies share a common assumption that the
CCNs are solely from a local source impacted by local boundary layer processes. Here we repeat
this type of CCN perturbation experiment and compare the resultant aerosol effects with our
current assessment for the effects of long-rang transported aerosols only. Three idealized CCN
profiles are used for the July 18 cases. The cloud susceptibility (ratio between logarithmic cloud
property change and logarithmic CCN change) derived from the comparison of those three
idealized runs are found to range from −0.22 to −0.25 for $R_e$ and from +0.18 to +0.30 for LWC
(Fig. 10a-b). Both $R_e$ and LWC susceptibility values are close to the high ends of the most of
current AIE assessments (Sato and Suzuki, 2018; Zheng et al., 2020). For the noticeable long-
range transport effect in the July 12 case, the $R_e$ and LWC susceptibilities are −0.11 and +0.14,
respectively. They are smaller than those from the idealized MBL aerosol perturbation experiments.
Hence, this suggests that the aerosols of long-range transport are less efficient in altering MBL
cloud properties than those originating from local sources. It can be attributed to the fact that dry
air likely enters cloud layer along with CCN, resulting in less supersaturation and reduced
activation rate.
**5. Conclusion and Discussion**
Located in the remote eastern North Atlantic, the Azores islands experience frequent long-
range transport of smoke and anthropogenic aerosols from continental U.S. A recent DOE ARM
ACE-ENA aircraft field campaign near the Azores in the summer of 2017 provides ample
observations of aerosols and clouds with detailed vertical information. In this study, we combine
the aircraft measurements, CAMS aerosol reanalysis, and an aerosol-aware and cloud-resolving
WRF model to characterize spatial variations of aerosols from long-range transport over the
Azores islands and assess their possible influence on the marine boundary layer clouds. The
reanalysis data show high frequency of occurrence of long-range transport over this area.
Evaluated by airborne aerosol measurement, the CAMS reanalysis data generally reproduce
observed aerosol profiles over this remote region, but the predicted aerosol mass mixing ratios are
still significantly biased. Our back-trajectory analyses confirm that anthropogenic and/or biomass
burning aerosols were mainly from the U.S. continent during the summer phase of ACE-ENA,
while the dust plumes are mainly originated from Sahara.
Aircraft observations show distinctive aerosol vertical distribution scenarios when long-



range transport of aerosols is noticeable. In some cases, a sharp decrease in aerosol concentration
downwards the cloud top with a minimal value right above the cloud top, while in other cases,
moderate decrease with a higher aerosol concentration near the cloud top than below the cloud
bottom. To identify the requirement for the long-range transported aerosols to exert significant
impacts on the MBL clouds beneath, a series of cloud resolving WRF simulations are conducted
for the selected cases. The model with dynamical downscaling from 19 km horizontal resolution
down to 300 m grid spacing is found reliable in simulating the vertical variability of temperature
and humidity fields over the Azores island, as well as in capturing the basic cloud structure. By
imposing aerosol plumes at the observed heights and varying them in the sensitivity runs, the
simulation results suggest the aerosol plume cannot affect underlying MBL cloud properties when
the center of the plume is over 100 m higher than cloud top. Even when the aerosols are right on
top of the stratified MBL cloud deck, the deepening of cloud and destabilization of boundary layer
are required to have significant aerosol-cloud interactions. We find more marine cloud fractions
with larger water content by the aerosols from long-range transport when the aerosol layer is
emerged into the cloud deck. For the case with noticeable long-range transport aerosol effect on
MBL cloud, the susceptibilities of droplet effective radius and liquid water content are $-0.11$ and
$+0.14$, respectively. Additional model sensitivity experiments are conducted, which perturb the
whole-column aerosol concentration without changing the shape of their vertical profiles. The
results show much larger susceptibility of cloud effective radius and liquid water path to the similar
magnitude of aerosol perturbation in PBL, indicating that the long-range transported aerosols are
less efficient in altering MBL cloud properties than those originated from local sources.
Through the comparisons of above- and below-cloud aerosol concentrations and the
examination of aerosol plume and cloud top height variations, we find about 63% occurrence
frequency of the interaction between remote aerosol and local MBL cloud based on the eight flights
during the summer phase of the ACE-ENA field campaign. Such a high frequency indicates the
importance of long-range transport aerosols on MBL clouds. Note that, due to the limited sample
size, the frequency may not be accurate to represent the true value on the daily basis. To our
knowledge, our study represents the first effort to utilize the ACE-ENA aircraft campaign data to
study the impacts of long-range transported aerosols on MBL clouds. Future study will focus on
the comparison of AIE involving long-range transport aerosols between different ARM sites and
field campaigns.





**Code availability**

The code of WRF model used in this study is available at https://www2.mmm.ucar.edu/wrf/users/downloads.html.


**Data availability**

All the WRF model simulation output used for this research can be downloaded from the website at http://web.gps.caltech.edu/~yzw/share/Wang-2020-ACP-Azores. The aircraft and ground-based measurements used in this study were obtained from the Atmospheric Radiation Measurement (ARM) Program sponsored by the U.S. Department of Energy (DOE) Office of Energy Research, Office of Health and Environmental Research, and Environmental Sciences Division. The data can be downloaded from http://www.archive.arm.gov/. CAMS global aerosol reanalysis product at pressure level used in this study can be downloaded at https://apps.ecmwf.int/datasets/data/cams-nrealtime/levtype=pl/. ERA5 data is available for download via the Copernicus Climate Data Store website (https://cds.climate.copernicus.eu).

**Acknowledgement**

This study was primarily supported by the collaborative NSF grant (Award No. AGS-1700727, 1700728). We acknowledge helpful discussions on the model setup with Dr. Zheng Lu at Texas A&M University. We thank the instrument mentors of the AMS, SP2, and CPC instruments and the individuals collecting measurements during the ACE-ENA field campaign. We also acknowledge high-performance computing support from Pleiades provided at NASA Ames. All requests for materials in this paper should be addressed to Yuan Wang (yuan.wang@caltech.edu).



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

**Figures**



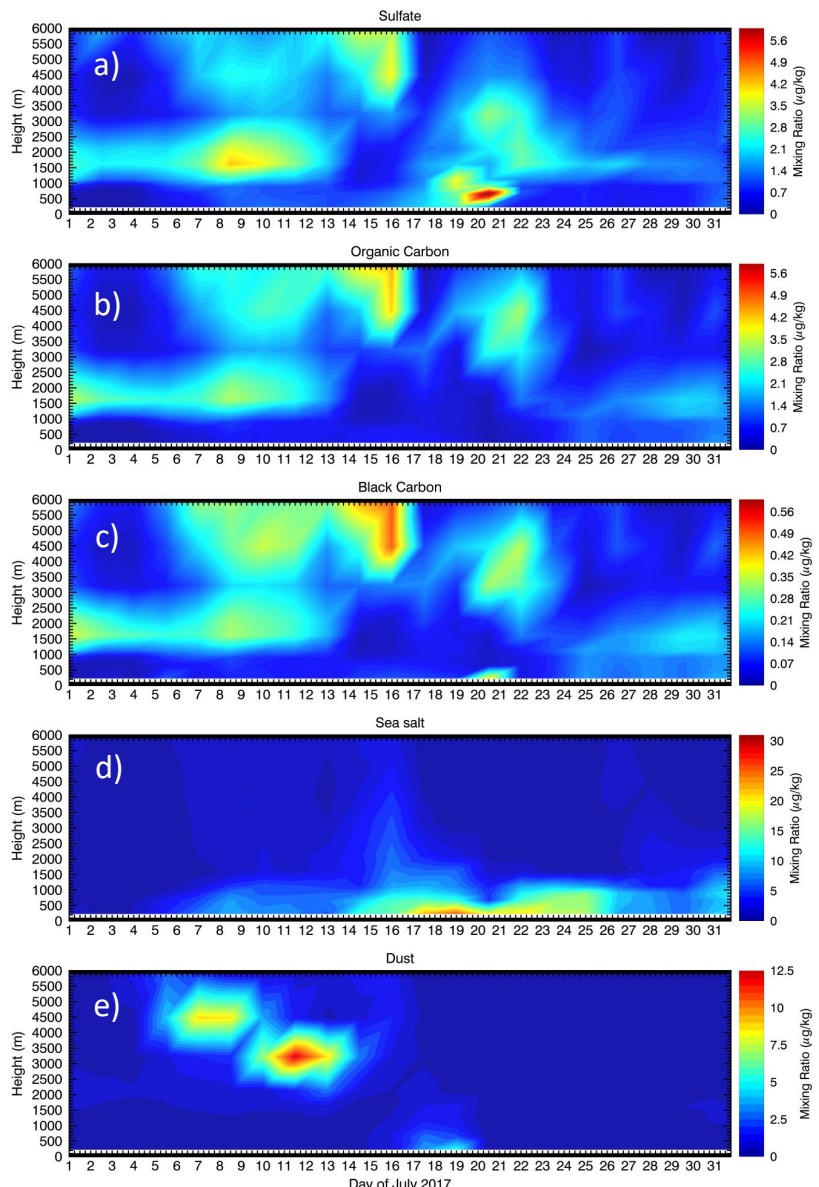

**Figure 1**. Temporal evolutions of vertical distributions for five types of aerosols as shown in a) sulfate, b) organic carbon, c) black carbon, d) sea salt, and e) dust during July 2017 over the Azores based on the ECMWF-CAMS aerosol reanalysis product.





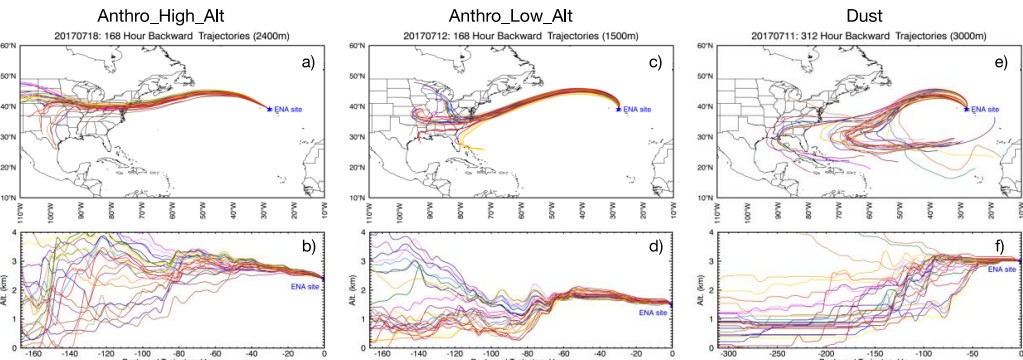

629

**Figure 2**. Back-trajectory analyses of airmass history starting from the ENA site for the three selected cases using the NOAA HYSPLIT Trajectory Model. Anthropogenic aerosols dominated plume with high altitude (Anthro_High_Alt) and low altitude (Anthro_High_Alt), dust plume (Dust).

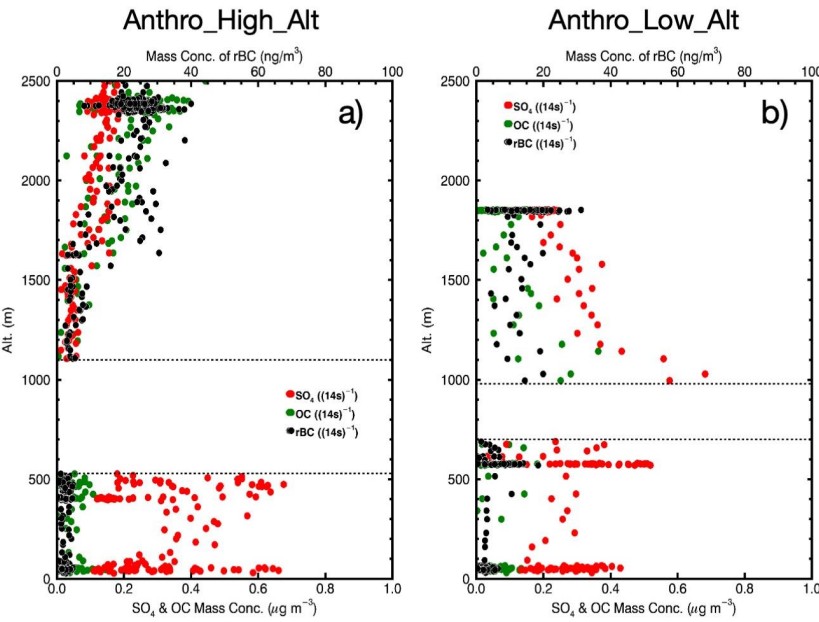

635

**Figure 3**. Airborne measured vertical profiles of sulfate (SO$_4$, red dots), organic carbon (OC, green

dots), and refractory BC (rBC, black dots) mass mixing ratios averaged over multiple flights in

two characteristic cases: (a) high-altitude aerosol plume on 18 July and (b) low-altitude aerosol

plume on 12 July, 2017. The highly uncertain and noisy aerosol observations due to cloud

contamination are not shown (between two dash lines), so the blank regions approximately denote

cloud layer.

642

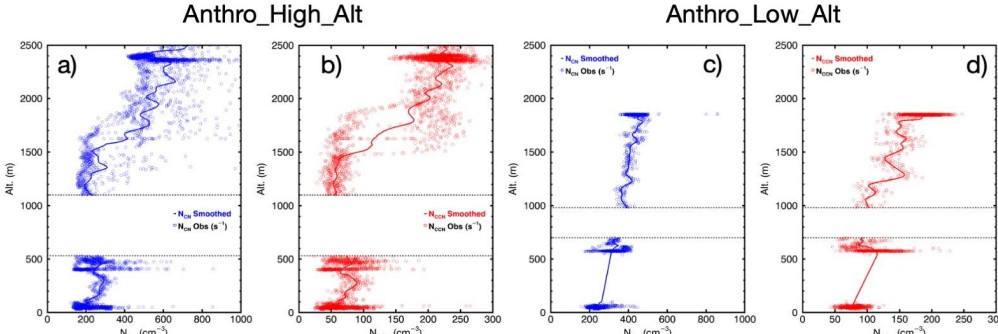

643

**Figure 4**. Airborne measured profiles of condensation nuclei ($N_{CN}$) and cloud condensation nuclei

($N_{CCN}$) averaged over multiple flights in two cases with high- and low-altitude aerosol plumes.

The highly uncertain and noisy aerosol observations due to cloud contamination are not shown

(between two dash lines), so the blank regions approximately denote cloud layer.






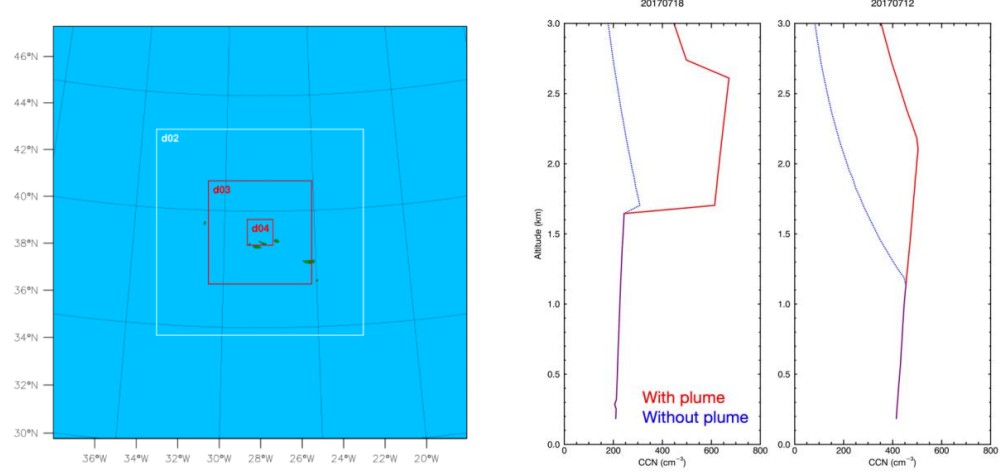


**Figure 5**. WRF domain map and aerosol concentration profiles used in the model as initial and
boundary conditions for the sensitivity runs of the two cases.





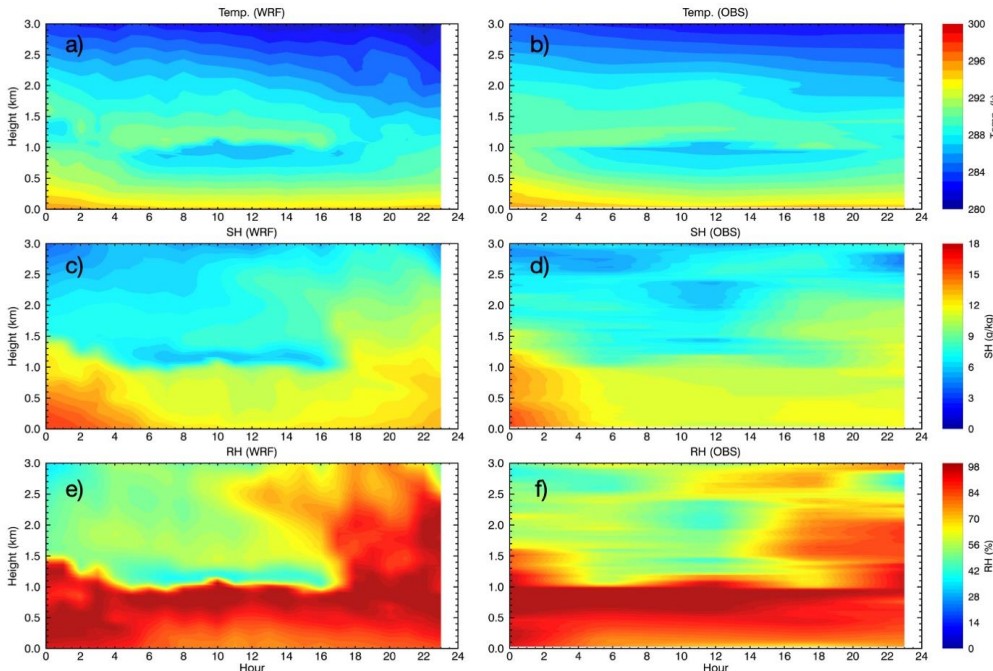


**Figure 6**. WRF simulated (left panels) and merged sounding measured (right panels) spatiotemporal evolutions of air temperature (the first row), specific humidity (the second row), and relative humidity (the third row) for the high-altitude plume case.






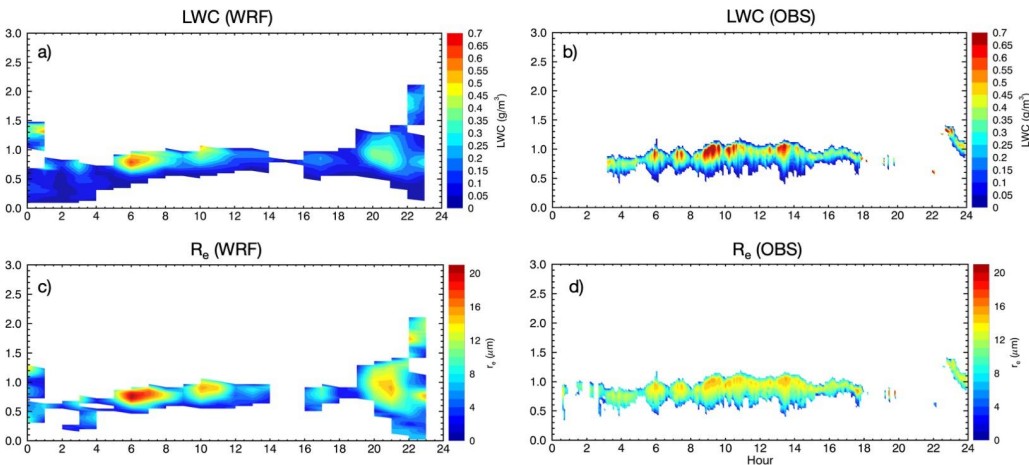


**Figure 7**. WRF simulated (top panels) and cloud radar retrieved (bottom panels) spatiotemporal
evolution of liquid water content (the left column) and droplet effective radius (the right column)
for the high-altitude plume case.



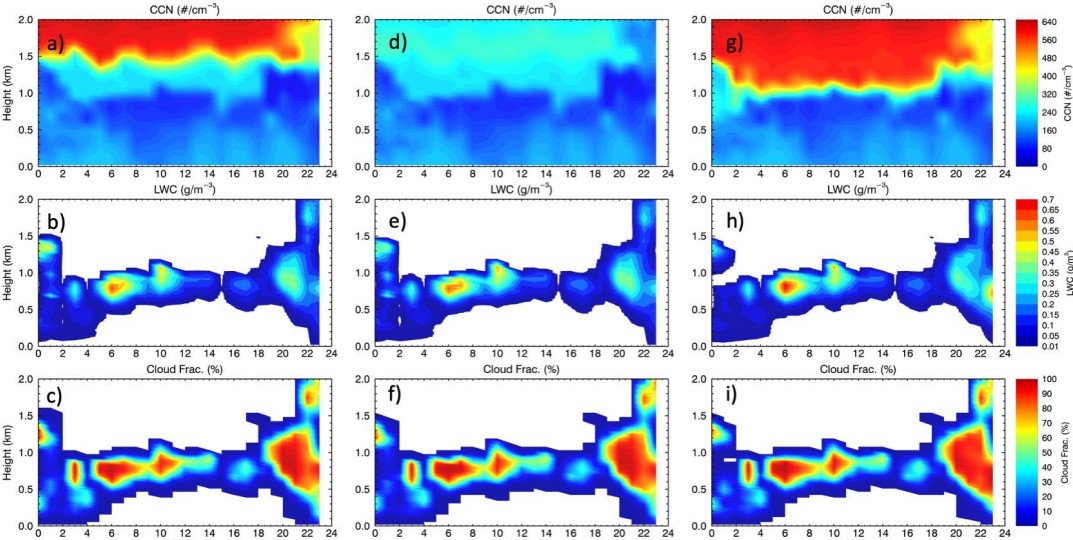


**Figure 8**. WRF simulated CCN concentration, liquid water content (LWC), and cloud fraction
for the high-altitude plume case (averaged over 20 × 20 grid points): a-c) with the observed
aerosol plume due to long-range transport (above 1.5 km), d-f) with the aerosol plume removed,
and g-i) with the aerosol plume moved downward to 1.1 km.






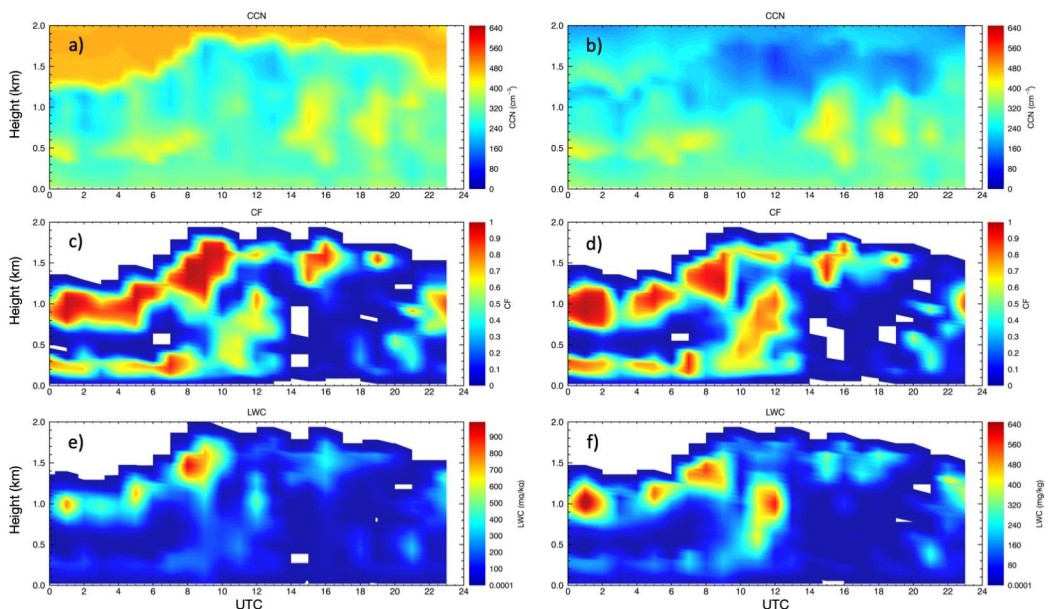


**Figure 9**. WRF simulated CCN concentration, liquid water content (LWC), and cloud fraction for
the low-altitude plume case.



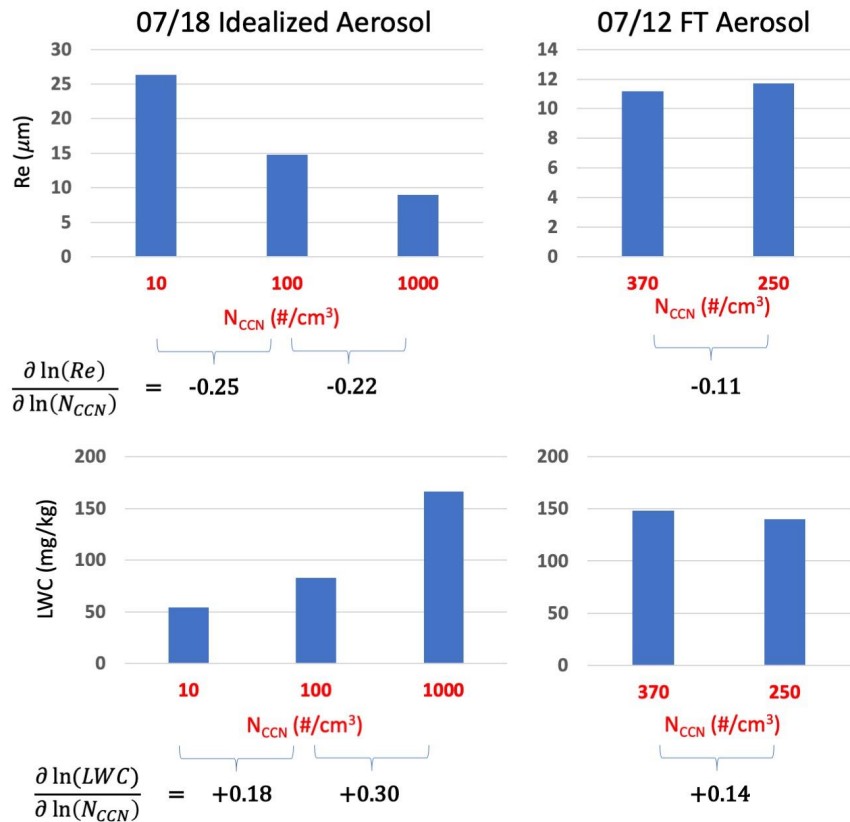


**Figure** 10. Model predicted cloud susceptibilities for the idealized CCN variations in the MBL

for the July 18 case and the influence of CCN variations in the free troposphere (FT) for the July

12 case. The cloud properties are averaged over all cloud points in the innermost domain. $N_{CCN}$

values are obtained from the initial CCN profiles and averaged over between 0.5-3 km.






**Table 1**. Characteristics of condensation nuclei concentration (CN)and cloud vertical profiles for
all eight cases during the summer phase of the DOE ACE-ENA field campaign.

| Date of Flight | Cloud Type | Above-Cloud Aerosol Changes with Height | Above-cloud $N_{CN}$* (# cm$^{-3}$) | Below-cloud $N_{CN}$* (# cm$^{-3}$) | Cloud Top Height Variation** (m) | Critical Altitude*** (m) |
|---|---|---|---|---|---|---|
| 20170628 | Thin Stratus | Increase | 471 | 353 | 670 - 1060 | N/A |
| 20170630 | Thin Stratus | Increase | 456 | 391 | 820 - 1270 | N/A |
| 20170706 | StCu. | Keep constant | 354 | 272 | 1210 - 1720 | 1820 |
| 20170707 | Stratus | Decrease | 266 | 247 | 1540 - 1960 | N/A |
| 20170712 | StCu. | Increase | 464 | 331 | 760 - 1360 | N/A |
| 20170715 | StCu. | Increase | 237 | 205 | 1120 - 1750 | N/A |
| 20170718 | StCu. | Increase | 185 | 290 | 880 - 1300 | 1674 |
| 20170720 | StCu. | Decrease | 224 | 311 | 970 - 1660 | N/A |

\* Average within 200 m of above (below) cloud top (base)
\*\* For continuous cloud layer
\*\*\* Critical altitude is defined as the height at which above-cloud $N_{CN}$ is equal to the below-
cloud $N_{CN}$.