# Peer review of "Impacts of Long-range Transport of Aerosols on Marine Boundary Layer Clouds in"

_Atmospheric Chemistry and Physics, 2020_

## Referee Comment (RC1) · Anonymous Referee #1 · 24 Aug 2020

This study characterizes the properties of long-range transport aerosols observed by analyzing in-situ measurements from the ACE-ENA field campaigns and ECMWF-CAMS aerosol reanalysis data. Cloud-resolving WRF simulations are used to assess the possible influence of long-range transport aerosols on marine boundary-layer clouds. Results show that long-range transport biomass burning aerosols from U.S. continent and dust plumes from Sahara are observed during the field campaign. In-situ measurements show that long-range transport aerosol layer is some distance away from the cloud top for one case and adjacent to the cloud top for another case. A series of WRF simulations suggest that the aerosol plume cannot affect underlying MBL cloud properties when the center of the plume is over 100 m higher than cloud top.

Noticeable effect of aerosol on cloud properties is found if the aerosol layer is right on top of the stratified MBL cloud deck. The manuscript is well written and the results and conclusions are clearly presented. I think the manuscript is suitable for publication in ACP after minor revision.

1. Line 188: "July 18 and 12 presents the typical high- and low-plume cases. . ." The signal is clear from Figure 3 (in-situ measurements), but not clear in Figure 1. In fact, based on Figure 1 (reanalysis product), I think July 18 is likely to be low-plume case, while July 12 is high-plume case. Please comment on the difference and add some explanations/clarifications in the manuscript.

2. Figure 6&7: Results are horizontally average in domain d04 or from one column where ENA site located? I guess it is averaged. Please clearly state it in the text and caption.

3. Figure 9: Caption is not completed. b), d), f) are case with the aerosol plume removed?

---

## Referee Comment (RC2) · Michael Diamond (Referee) · 29 Aug 2020

In their manuscript, Wang et al. compare observed vertical profiles of aerosols at the Azores from the July 2017 flight campaign around the ACE-ENA ground site with re-analysis data from CAMS and ERA5 and with new simulations from nested WRF-AAM simulations. The WRF simulations span from "regional model" ($\sim$20 km) to "cloud resolving" (300 m horizontal) resolution. Observed and reanalysis aerosol distributions match qualitatively. It is shown that variations in both aerosol plume bottom and cloud top height matter for whether and how much aerosol is actually entrained into the marine boundary layer. The authors conclude that MBL aerosol variations influence cloud

[Figure]
* * *
Interactive
comment

properties more than free troposphere variations. I agree with this conclusion overall but have some questions about their interpretation of the idealized experiments in Figure 10. That notwithstanding, the manuscript is well-organized, clearly written, and presents interesting new results. I recommend publication following minor revisions.

Major comments:

1. It would be helpful to keep in mind that the observations only show instantaneous contact between aerosol in the free troposphere (from long-range transport) and MBL. Although the MBL concentrations seemingly are assumed to be from local sources, it is also possible that there is a contribution from FT aerosol entrained earlier and transported with the MBL flow. Instantaneous snapshots of FT aerosol-MBL top contact (or lack thereof) cannot capture the effects of previously entrained aerosol.

2. Page 14, Lines 378-389: If the CCN perturbation is being averaged between 500 m and 3 km (as stated in the Figure 10 caption), then a lot of the CCN in the elevated plume case is irrelevant to the cloud properties. The lower susceptibility values are those an artifact of averaging in aerosol that isn't doing anything to affect the clouds. This is consistent with the interpretation of different above-cloud and below-cloud susceptibility values in Diamond et al. (2018). I'm not convinced this says anything in particular about aerosol source *once that aerosol is in the MBL*. (E.g., the difference between 500 particles/mg from a local source versus the entrainment of 500 particles/mg in the FT from long-range transport, assuming the same initial MBL background concentration.) The conclusions as written strike me as being overly broad for the evidence presented.

Specific comments:

1. Page 2, Line 16: No evidence in the text is provided about the accuracy of the instrumentation, so this descriptor probably doesn't belong in the abstract.

2. Page 2, Line 26: Why "preferably"? Aerosol near the MBL top is a necessary

condition for entrainment and thus influence on indirect effects. Perhaps an argument can be made that absorbing aerosol well-separated from the MBL could have important semi-direct effects, but that's not addressed in the paper.

3. Page 4, Line 64: There is similar work to Diamond et al. (2018) looking at several aircraft campaigns based out of California (Mardi et al., 2019). This may be worth mentioning as the aerosol concentrations typical of the Azores are likely more similar to the northeast Pacific than to the southeast Atlantic during seasons with extensive biomass burning aerosol plumes.

Mardi, A. H., Dadashazar, H., MacDonald, A. B., Braun, R. A., Crosbie, E., Coggon, M. M., et al. (2019). Effects of Biomass Burning on Stratocumulus Droplet Characteristics, Drizzle Rate, and Composition. Journal of Geophysical Research: Atmospheres, 124.

4. Page 4, Lines 69-70: There are two distinct issues that may be getting blurred here: satellites missing thin aerosol layers (what is addressed explicitly) and satellites saturating and underestimating the extent of thick layers (e.g., Rajapakshe et al., 2017).

Rajapakshe, C., Zhang, Z., Yorks, J. E., Yu, H., Tan, Q., Meyer, K., et al. (2017). Seasonally transported aerosol layers over southeast Atlantic are closer to underlying clouds than previously reported. Geophysical Research Letters, 44(11), 5818-5825.

5. Page 5, Line 104: Is there any relevant literature you can cite for the accuracy of the measurements during the campaign? They are used as "truth" and not evaluated directly in the present paper.

6. Page 6, Line 141: There is no supporting information I could find. Did you mean to reference the map in Figure 5 here?

7. Page 11, Line 308: You may want to consider adding an in-text or supporting information figure here showcasing the improvement when using the outmost domain. This discussion seems very useful for others interested in performing similar modeling work and could probably be highlighted a bit more.

8. Page 12, Lines 331-333: I would urge some caution in the comparison with Painemal et al. (2014), as that paper's results may have been influenced by the low bias in CALIOP-derived plume base altitude discussed earlier (Rajapakshe et al., 2017) and the authors do discuss this issue as well.

9. Page 13, Line 347: Where, vertically, is the CCN that is being quantified here? I was interpreting this as an MBL average, but it would be helpful to be explicit here.

10. Page 15, Line 415: Isn't it the bottom of the FT plume, not its "center", that should matter for the discussion here? One can easily imagine a very thick plume (like in the southeast Atlantic) that interacts with the cloud top significant but is "centered" at much higher altitude.

11. Figure 9: The caption does not seem to describe the entire figure. It should more fully explain the differences between the two columns.

Technical comments:

1. Page 5, Line 124: "Were" instead of "are"? I believe you are referring to the previously published results of Wu et al. (2020) to justify the assertion here, but the current phrasing makes it sound like this work is performed in the present paper.

2. Page 8, Line 206: "However" instead of "either"? I don't understand what the "either" would be referring to.
* * *

---

## Author Comment (AC1) · 5 Oct 2020

**Referee 1:**

This study characterizes the properties of long-range transport aerosols observed by analyzing in-situ measurements from the ACE-ENA field campaigns and ECMWFCAMS aerosol reanalysis data. Cloud-resolving WRF simulations are used to assess the possible influence of long-range transport aerosols on marine boundary-layer clouds. Results show that long-range transport biomass burning aerosols from U.S. continent and dust plumes from Sahara are observed during the field campaign. In situ measurements show that long-range transport aerosol layer is some distance away from the cloud top for one case and adjacent to the cloud top for another case. A series of WRF simulations suggest that the aerosol plume cannot affect underlying MBL cloud properties when the center of the plume is over 100 m higher than cloud top.

Noticeable effect of aerosol on cloud properties is found if the aerosol layer is right on top of the stratified MBL cloud deck. The manuscript is well written and the results and conclusions are clearly presented. I think the manuscript is suitable for publication in ACP after minor revision.

We appreciate the reviewer's valuable comments and constructive suggestions. We have carefully revised the manuscript according to these comments. Point-by-point responses are provided below. The reviewer's comments are in black, our responses are in blue, and the quotes from our manuscript are in italics.

1. Line 188: "July 18 and 12 presents the typical high- and low-plume cases. . ." The signal is clear from Figure 3 (in-situ measurements), but not clear in Figure 1. In fact, based on Figure 1 (reanalysis product), I think July 18 is likely to be low-plume case, while July 12 is high-plume case. Please comment on the difference and add some explanations/clarifications in the manuscript.

We have now clarified that sulfate occurrence below 1 km during 18-21 July in the reanalysis is unlikely caused by the long-range transport. The sulfate concentration experienced an increase in the MBL followed by a lag increase in the free troposphere. The elevated sulfate concentration within the boundary layer is due likely to some local sources such as oxidation of marine dimethyl sulfate (DMS) in the CAMS model. Also, the aircraft did not detect such a sulfate enhancement within the boundary layer on 18 July. Therefore, the July 18 is still considered as a high-plume case in this study.

2. Figure 6&7: Results are horizontally average in domain d04 or from one column where ENA site located? I guess it is averaged. Please clearly state it in the text and caption.

In the captions of Fig. 6&7, we have now added that "The model results are averaged over 10×10 grid points centering at the ENA ground site location".

3. Figure 9: Caption is not completed. b), d), f) are case with the aerosol plume removed?

We have completed the caption of Fig. 9 as "*WRF simulated CCN concentration, liquid water content (LWC), and cloud fraction for the low-altitude plume case, with observed aerosol profile (a,c,e) and idealized profile that removed aerosol transport in the free troposphere (b,d,f).*"

---

## Author Comment (AC2) · 5 Oct 2020

**Referee 2:**

In their manuscript, Wang et al. compare observed vertical profiles of aerosols at the Azores from the July 2017 flight campaign around the ACE-ENA ground site with reanalysis data from CAMS and ERA5 and with new simulations from nested WRF-AAM simulations. The WRF simulations span from "regional model" (20 km) to "cloud resolving" (300 m horizontal) resolution. Observed and reanalysis aerosol distributions match qualitatively. It is shown that variations in both aerosol plume bottom and cloud top height matter for whether and how much aerosol is actually entrained into the marine boundary layer. The authors conclude that MBL aerosol variations influence cloud properties more than free troposphere variations. I agree with this conclusion overall but have some questions about their interpretation of the idealized experiments in Figure 10. That notwithstanding, the manuscript is well-organized, clearly written, and presents interesting new results. I recommend publication following minor revisions.

We appreciate the reviewer's valuable comments and constructive suggestions. We have carefully revised the manuscript according to these comments. Point-by-point responses are provided below. The reviewer's comments are in black, our responses are in blue, and the quotes from our manuscript are in italics.

Major comments:
1. It would be helpful to keep in mind that the observations only show instantaneous contact between aerosol in the free troposphere (from long-range transport) and MBL. Although the MBL concentrations seemingly are assumed to be from local sources, it is also possible that there is a contribution from FT aerosol entrained earlier and transported with the MBL flow. Instantaneous snapshots of FT aerosol-MBL top contact (or lack thereof) cannot capture the effects of previously entrained aerosol.

The reviewer made a good point here. This is also the reason why we did not simply compare $N_{CN}$ below and above cloud layer and take further step to examine their vertical variations (increasing or decreasing with height) above cloud top as well as the variations of cloud top height. As suggested, the uncertainty of aircraft observations has now been discussed on Page 15: "*Note that in situ observations only show instantaneous conditions of aerosols in the free troposphere and MBL, and they are subject to the influence from earlier aerosol entrainment or horizontal transports with the MBL flow*".

2. Page 14, Lines 378-389: If the CCN perturbation is being averaged between 500 m and 3 km (as stated in the Figure 10 caption), then a lot of the CCN in the elevated plume case is irrelevant to the cloud properties. The lower susceptibility values are those an artifact of averaging in aerosol that isn't doing anything to affect the clouds. This is consistent with the interpretation of different above-cloud and below-cloud susceptibility values in Diamond et al. (2018). I'm not convinced this says anything in particular about aerosol source *once that aerosol is in the MBL*. (E.g., the difference between 500 particles/mg from a local source versus the entrainment of 500 particles/mg in the FT from long-range transport, assuming the same initial MBL background concentration.) The conclusions as written strike me as being overly broad for the evidence presented.

We agree with the reviewer that it is difficult to accurately pinpoint the aerosols involved in aerosol-cloud interactions, considering the possible aerosol exchange between MBL and FT. The rationale in our paper to average over multiple levels between 500 and 3000 m is to include all possible relevant aerosols and facilitate a fair comparison of two aerosol scenarios with distinctive profiles, i.e. one mainly in the MBL (0.5-1.5 km) and the other mainly in the FT (1-3 km). Those two scenarios are defined not by where we see the aerosols during the simulations, but by the initial profiles we imposed. For the cases with aerosols originated from MBL, we use three bottom-heavy profiles (well-mixed in MBL and exponentially decreasing in FT) as the initial conditions for CCN. The CCN concentrations in MBL are 10, 100, and 1000 cm$^{-3}$ in three sensitivity runs. We have now clarified that the cloud

susceptibility in our analysis is defined as the ratio between logarithmic cloud property changes in the simulations and logarithmic CCN differences in the initial profiles.

We acknowledge that averaging CCN over broad spatial range may introduce uncertainty to the susceptibility quantification. However, the purpose of this analysis is to compare the relative importance of aerosols in different levels, rather than the calculation of absolute values of cloud susceptibility. We have now discussed the caveat of this analysis method on page 15.

Specific comments:
1. Page 2, Line 16: No evidence in the text is provided about the accuracy of the instrumentation, so this descriptor probably doesn't belong in the abstract.

The word "accurate" has been removed.

2. Page 2, Line 26: Why "preferably"? Aerosol near the MBL top is a necessary condition for entrainment and thus influence on indirect effects. Perhaps an argument can be made that absorbing aerosol well-separated from the MBL could have important semi-direct effects, but that's not addressed in the paper.

The word "preferably" has been removed and add "plume" after aerosol. We meant to say aerosol plume should get close to the cloud deck, and the plume here means the majority of aerosols. Some aerosol may settle downward and touch the cloud, but it doesn't necessarily require the majority of the aerosols in that plume behave the same.

3. Page 4, Line 64: There is similar work to Diamond et al. (2018) looking at several aircraft campaigns based out of California (Mardi et al., 2019). This may be worth mentioning as the aerosol concentrations typical of the Azores are likely more similar to the northeast Pacific than to the southeast Atlantic during seasons with extensive biomass burning aerosol plumes.
Mardi, A. H., Dadashazar, H., MacDonald, A. B., Braun, R. A., Crosbie, E., Coggon, M. M., et al. (2019). Effects of Biomass Burning on Stratocumulus Droplet Characteristics, Drizzle Rate, and Composition. Journal of Geophysical Research: Atmospheres, 124.

As suggested, we have now added the discussion as "*in the northeast Pacific where aerosol types are similar with the Azores, biomass burning aerosols from the episodic wildfire events are found less efficient in altering cloud microphysics than the non-biomass burning aerosols (Mardi et al., 2019)*".

4. Page 4, Lines 69-70: There are two distinct issues that may be getting blurred here: satellites missing thin aerosol layers (what is addressed explicitly) and satellites saturating and underestimating the extent of thick layers (e.g., Rajapakshe et al., 2017).
Rajapakshe, C., Zhang, Z., Yorks, J. E., Yu, H., Tan, Q., Meyer, K., et al. (2017). Seasonally transported aerosol layers over southeast Atlantic are closer to underlying clouds than previously reported. Geophysical Research Letters, 44(11), 5818-5825.

we have now discussed those two issues separately by adding "*Also, when plumes are too thick near the aerosol source regions, satellite signals will be saturated and the retrievals may underestimate the extent of thick layers (Rajapakshe et al., 2017)*".

5. Page 5, Line 104: Is there any relevant literature you can cite for the accuracy of the measurements during the campaign? They are used as "truth" and not evaluated directly in the present paper.

We haven't found any literature on the product validation during the ACE-ENA campaign. The accuracy of each individual instrument can be found in the instrument handbooks available at the ARM website. We have clarified it in the text.

6. Page 6, Line 141: There is no supporting information I could find. Did you mean to reference the map in Figure 5 here?

The typo has been fixed.

7. Page 11, Line 308: You may want to consider adding an in-text or supporting information figure here showcasing the improvement when using the outmost domain. This discussion seems very useful for others interested in performing similar modeling work and could probably be highlighted a bit more.

As suggested, we have now added three new panels in Figure 6 to show the model sensitivity simulation without the outmost domain and to illustrate the importance of the large-scale forcing on the MBL cloud structure.

[Figure]

**New Figure 6**.

8. Page 12, Lines 331-333: I would urge some caution in the comparison with Painemal et al. (2014), as that paper's results may have been influenced by the low bias in CALIOP-derived plume base altitude discussed earlier (Rajapakshe et al., 2017) and the authors do discuss this issue as well.

The comparison with Painemal et al. (2014) is removed now. Instead, we state "*This finding echoes the importance of accurate detection of plume base altitude using the remote sensing instruments (Rajapakshe et al., 2017)*".

9. Page 13, Line 347: Where, vertically, is the CCN that is being quantified here? I was interpreting this as an MBL average, but it would be helpful to be explicit here.

We have now clarified that the reported CCN changes occur between 500 and 3000 m in altitude.

10. Page 15, Line 415: Isn't it the bottom of the FT plume, not its "center", that should matter for the discussion here? One can easily imagine a very thick plume (like in the southeast Atlantic) that interacts with the cloud top significant but is "centered" at much higher altitude.

It should be the "bottom" instead of "center". Revised.

11. Figure 9: The caption does not seem to describe the entire figure. It should more fully explain the differences between the two columns.

Now we have revised the Fig. 9 caption as "*WRF simulated CCN concentration, liquid water content (LWC), and cloud fraction from the low-altitude plume case, with observed aerosol profile (a,c,e) and idealized profile that removes aerosol transport in the free troposphere (b,d,f). The two different vertical profiles are shown in Fig. 5*".

Technical comments:
1. Page 5, Line 124: "Were" instead of "are"? I believe you are referring to the previously published results of Wu et al. (2020) to justify the assertion here, but the current phrasing makes it sound like this work is performed in the present paper.

Changed to "were" as suggested.

2. Page 8, Line 206: "However" instead of "either"? I don't understand what the "either" would be referring to.

Changed to "however" as suggested.